## Research Article

peer support; recovery; global mental health; UPSIDES; qualitative research; low- and middle-income countries

**Corresponding authors:**
Galia S Moran and Yael Goldfarb,
Emails: galiam@bgu.ac.il; goldfarbya@gmail.com.

# Experiences of service users receiving peer support in mental health services: Qualitative findings from the international UPSIDES trial

Yael Goldfarb[1,2] ⬤, Alina Grayzman[1], Sarah Barber[3], Cerdic Hall[4], Maria Haun[5], Jasmine Kalha[6], Silvia Krumm[5,7], Rachel Mtei[8], Rebecca Nixdorf[9], Bernd Puschner[5] ⬤, Mike Slade[10,11] and Galia S. Moran[1]

[1]Social Work, Ben-Gurion University of the Negev, Israel; [2]Faculty of Business Administration, College of Law and Business Ramat Gan, Israel; [3]Butabika National Referral Mental Hospital, Uganda; [4]East London NHS Foundation Trust, UK; [5]Ulm University Medical Center Department of Psychiatry and Psychotherapy II, Germany; [6]Centre for Mental Health Law and Policy, Indian Law Society, India; [7]Leipzig University, Germany; [8]Ifakara Health Institute, United Republic of Tanzania; [9]University Medical Center Hamburg-Eppendorf, Germany; [10]University of Nottingham, UK and [11]Nord University – Namsos, Norway

## Abstract

Although peer support is increasingly used in mental health services worldwide, service users' experiences have been studied mostly in high-income countries. The current study examined service users' experiences of peer support in the UPSIDES Trial, delivered across diverse cultural and resource contexts, including high, middle and low-income countries. Semi-structured interviews were conducted with 33 service users across six study sites (Germany [two sites], Uganda, Tanzania, Israel, and India) and analyzed using thematic analysis to identify patterns in participants' experiences. To capture diverse perspectives, service users were purposively sampled based on pre–post changes in social inclusion and personal recovery, with participants randomly selected from the top and bottom 20% ('high' and 'low' responders). Four themes emerged: (1) adaptable settings and intervention flexibility; (2) 'active ingredients' such as mutuality, reciprocity, and role-modeling; (3) positive intra-personal, inter-personal, and behavioral outcomes; and (4) barriers, including mismatches, unmet expectations, unclear boundaries and challenges to continuity. The study highlights shared relational elements of peer support alongside context-specific adaptations. Findings reinforce its value as a complementary, person-centred service with global relevance, while pointing to challenges including improving matching, reducing dropout, and clarifying expectations. Site-specific aspects are discussed, offering insights for global implementation.

## Impact Statements

Understanding how peer support is experienced across high-, middle-, and low-income settings is crucial for strengthening recovery-oriented mental health care globally. This study provides cross-site findings from the UPSIDES Trial, presenting the core mechanisms of peer support that matter to service users, while highlighting the importance of flexibility in responding to individual, cultural, and resource-related needs. By identifying practical challenges such as matching, unmet expectations, and discontinuity, the findings offer guidance for policymakers, programme developers, and clinical leaders seeking to implement or scale up peer-support services. Grounded in diverse service-user experiences, findings shed light on ways to promote more adaptable, person-centred, and contextually responsive mental health systems worldwide.

## Introduction

Mental health peer support (MHPS) services involve individuals with lived mental health experiences supporting others with similar challenges. Recognized as a valuable addition to traditional mental health care, MHPS has been linked to improved outcomes in various global settings, including reduced depression and anxiety symptoms, increased empowerment, and better social functioning (Walker and Bryant, 2013; Lyons et al., 2021; Mutschler et al., 2022; Smit et al., 2023; Asher et al., 2024; Chow et al., 2025). As a result, MHPS is increasingly integrated into healthcare settings, highlighting its role in promoting recovery (Moran, 2025; Moran et al., 2025). Service user perspectives have been relatively under-researched compared to those of peer support workers (PSWs).

Studies that include service user perspectives show benefits such as building more trusting, recovery-oriented relationships with healthcare providers, offering inspiration and practical

recovery strategies, serving as positive role models, reducing self-stigma, and strengthening community belonging (Gillard et al., 2015; Tse et al., 2017; Rosenberg and Argentzell, 2018; Hiller-Venegas et al., 2022; Marks et al., 2022).

Along with the benefits, other studies have identified challenges and factors that may impede the implementation of MHPS. Some suggest that effectiveness may decrease because of relational factors, such as mismatches between PSWs and service users, or concerns that instability in a PSW's own recovery can trigger negative emotions or hinder effective support (Le et al., 2022). Role ambiguity, often a challenge raised by PSWs, also affected service users, leading to confusion and mistrust in the relationship (Lewis and Foye, 2022; Ong et al., 2023). Furthermore, overly rigid approaches from PSWs, such as enforcing their own recovery paths, were counterproductive and undermined the service users' autonomy (Ogundipe et al., 2019). Finally, institutional factors such as strict regulations (*e.g.*, restricting home visits, limiting public interaction) can undermine flexibility, which is central to the effectiveness of MHPS (Ogundipe et al., 2019). While these growing efforts to expand understandings of service user experiences lead to valuable insights, most were conducted within single-country contexts, commonly high-income countries. There is a limited understanding of how such experiences unfold across different cultural, organizational, and resource settings such as middle and low-income countries. Mental health systems, cultural understandings of recovery, stigma, and expectations regarding support relationships may vary considerably across countries. Examining peer support across diverse contexts may therefore help identify both shared experiential mechanisms and context-specific influences.

The current study aims to address this gap by examining service users' experiences of the UPSIDES peer support intervention across diverse socio-economic, cultural and organizational contexts, including high, middle and low-income countries (Moran et al., 2020). Exploring perspectives from these varied settings may provide deeper insight into how peer support is experienced, including perceived mechanisms of impact, conditions for optimal experiences, and barriers (Curry et al., 2009).

## Method

### The context of the study

The study was conducted as part of UPSIDES (Using Peer Support in Developing Empowering Mental Health Services) trial, a multinational research initiative aimed at developing sustainable best practice of MHPS in high-, middle- and low-resource settings (Moran et al., 2020). Six study sites participated in the trial: Hamburg and Ulm/Guenzburg (Germany), Butabika (Uganda), Dar es Salaam (Tanzania), Be'er Sheva (Israel), Pune (India). Sites were embedded in diverse mental health care systems and cultural contexts. In Tanzania, Uganda and India for example, services were largely hospital-based, while in Israel the intervention was delivered in the setting of a community psychiatric rehabilitation system. In relation to previous experience of peer support within the mental-health system, some sites, such as Hamburg (Germany) had prior experience with structured training programs and involvement of peer workers within mental health services, while in other sites peer support was newly introduced (*e.g.* Ulm-Germany and Dar es Salaam-Tanzania). See Supplementary File S2 for detailed site descriptions of site specific context.

### Development

The UPSIDES intervention was developed through a comprehensive process involving all participating countries, including workshops and consultations with service users, peer workers, clinicians and other stakeholders, to ensure cultural and contextual flexibility while maintaining fidelity to shared core principles of peer support being: mutual, reciprocal, non-directive, strength-based, and recovery- and community-oriented (HIltensperger et al., 2024; Nixdorf et al., 2024).

### Training and implementation

UPSIDES PSWs participated in a shared train-the-trainer workshop and subsequently delivered the intervention at their respective sites. PSWs were integrated within mental health services and teams (Moran et al., 2025). They received compensation for their work, with payment arrangements varying across study sites (*e.g.*, salaried positions or per-day payments), and structured support provided through supervision and local implementation practices (Puschner et al., 2025). Intended duration of the intervention was up to 6 months, with a minimum of three contacts between PSW and service user. Weekly or biweekly meetings were recommended, but frequency could vary. Meetings took place either one-on-one, in a small group format (Goldfarb et al., 2024), or a combination of both. The content of the meetings was dynamic and dialogical, adhering to the principles mentioned above.

### Evaluation

The intervention was evaluated through a pragmatic multicentre randomized controlled trial, with pre, post and several follow up assessments. The primary outcome assessed was social inclusion. Empowerment, hope, recovery and health and social functioning were assessed as secondary outcomes (Moran et al., 2020; Puschner et al., 2025). Results in the full trial sample indicated beneficial effects of the intervention on social inclusion as well as recovery-related outcomes including empowerment and hope (Puschner et al., 2025). In addition to the randomized controlled trial, the UPSIDES study incorporated qualitative components exploring implementation and experiential dimensions of peer support across sites (Haun et al., 2024; Moran et al., 2025), including the current study.

### Participants

33 participants were interviewed. Inclusion criteria for the intervention were: age 18–60; main diagnosis of a mental disorder; severe mental illness (Threshold Assessment Grid(Rosen et al., 2000) ≥5 points and illness duration ≥2 years); sufficient command of the host country's language; and capability of giving informed consent. Exclusion criteria: learning disabilities, dementia, substance use or organic brain disorders; and severe cognitive impairment affecting consent or study participation.

### Procedure

Following the intervention, service users were purposively selected at each site to capture a broad spectrum of responses. Service users were grouped as 'low responders' (least benefit) and 'high responders' (most benefit) based on pre-post intervention change scores from quantitative measures of social inclusion (Social Inclusion Scale; Secker et al., 2009) and service users' experiences of support for recovery received from their peer support worker (Brief INSPIRE; Williams et al., 2015). The top and bottom 20% of a

combined ranking were randomly selected and approached until a sufficient number of participants from each site were interviewed (pre-determined to be between 6 and 8), or when all listed service users were approached. Documented reasons for not participating were a decline in mental or physical health, difficulties attaining consent due to mental illness, and geographical relocation. All participants included in the qualitative sample had completed the intervention.

Interviews were conducted by trained research staff at each study site. These local researchers were members of the UPSIDES research teams and were involved in the broader project, but they were not responsible for delivering the peer support intervention.

The interviews followed a flexible interview guide that included 6–8 open questions addressing participants' experiences of peer support, perceived impacts on recovery and daily life, contextual facilitators and barriers to participation, the influence of COVID-19, and overall reflections on the intervention. The interview guide was developed collaboratively, standardized across sites, and systematically translated to support cultural and conceptual equivalence (Charles et al., 2022). The topic guide is available in Supplementary File S3. All research staff received joint guidance to support consistent implementation across sites.

Interviews were audio recorded and transcribed verbatim in the local language. Transcripts were back checked by the moderator against the recording for accuracy. Personal information in the transcripts was deleted or anonymised. To ensure consistent analysis across sites, a bilingual speaker recruited by the UPSIDES team translated the transcripts and field notes from the local language into English before finalization. Two researchers, part of the UPSIDES translation team, checked all translated transcripts to ensure comprehensibility for analysis (Charles et al., 2022). The study is reported in accordance with the Consolidated Criteria for Reporting Qualitative Research (COREQ; Tong et al., 2007). The completed checklist is provided in Supplementary File S1.

All procedures were conducted in accordance with the ethical standards of the institutional and national research committees. Approvals were obtained from the relevant committees in Germany, Uganda, Tanzania, Israel, and India. Written informed consent was obtained from all participants.

### Data analysis

The transcripts were analyzed with MAXQDA 24 software, employing thematic analysis following these steps (Terry et al., 2017): (1) familiarization and initial coding – all interview transcripts and field notes were read by two researchers (YG & AG), who suggested initial codes and themes inductively from the data. Memos captured preliminary thoughts; (2) theme development – initial codes were clustered into meaningful patterns as themes, guided by the research questions and aims; (3) reviewing and defining themes – researchers conducted parallel coding on three transcripts, compared results, resolved inconsistencies, and ensured codes aligned with the central organizing concept of each theme. These were then discussed with a third researcher (GM). The remaining data were subsequently coded, ensuring themes captured all relevant content. The research team considered whether experiential patterns differed between these groups during coding and theme development. Notably, although the sampling strategy targeted 'high' and 'low' responders to capture distinct experiences, content analysis did not reveal

consistent distinctions between these groups; (4) cross-site validation: themes were validated with research workers at each site to ensure perspectives from all participating contexts informed interpretation of the data.

## Results

Participants ranged in age from 20 to 59 years (M = 37.91, SD = 11.26). A majority of participants were not employed, and many were living alone. See Table 1 for full details.

Four themes describing service user experience of the UPSIDES peer support intervention were identified. Representative quotes are presented in Table 2.

### Theme 1: Adaptable settings and intervention flexibility

UPSIDES was implemented across diverse cultures and contexts. Participants described variations in aspects such as: one-on-one *versus* group meetings, the number and frequency of sessions, communication mode (*e.g.*, in-person, telephone, WhatsApp),

**Table 1.** Participants' socio-demographic characteristics

| Variable | Category | | |
|---|---|---|---|
| Age | Average (years) | – | ~38 |
| | Range (years) | – | 20–59 |
| | | **Frequency** | **Percentage** |
| Study Site | Be'er Sheva (BGU) | 6 | 18% |
| | Butabika (BU) | 5 | 15% |
| | Dar es Salaam (DS) | 5 | 15% |
| | Pune (PU) | 4 | 12% |
| | Hamburg (UKE) | 7 | 21% |
| | Ulm (ULM) | 6 | 18% |
| Gender | Male | 15 | 45% |
| | Female | 18 | 55% |
| Personal Status | Single | 20 | 61% |
| | Married | 7 | 21% |
| | Divorced | 4 | 12% |
| | Separated | 2 | 6% |
| Living Situation | Living alone (+/− children) | 16 | 49% |
| | Living together as a couple | 8 | 24% |
| | Living with parents/ family member | 6 | 18% |
| | Living with spouse (+/− children) | 3 | 9% |
| Employment status | Unemployed | 20 | 61% |
| | Paid or self-employment | 4 | 12% |
| | Retired | 3 | 9% |
| | Sheltered employment | 2 | 6% |
| | Other | 4 | 12% |

**Table 2.** Key themes and supporting quotes

| Theme | Quotes |
|---|---|
| *Theme 1: Adaptable settings and intervention flexibility* | I would get him a chair and we would sit at the veranda…he used to come before afternoon at mid mornings, and that was convenient for me because he would find [me] when I am done with having a shower and breakfast" (BU70, 26, male). |
| | We met quite often in the city and often visited different places that are also interesting to visit, such as a lake… we played table tennis… once we also visited a climbing hall … I thought that was great too. That you can get out a bit and be a bit distracted (ULM117, 23, male). |
| | There were appointments where we saw each other two weeks in a row and then there were appointments where I canceled for health reasons - she never canceled - I was the one who canceled several times for health reasons… either four, five or six weeks we did not hear each other at all… she only did one thing, which I thought was very good she always, always offered [to make a phone appointment] and we have. She always tried to make any appointments possible (UKE65, 52, female) |
| | It was good for me to be out in the fresh air, walking along forest paths or looking for a quiet park bench, the weather like that … so, Corona did not really affect me (ULM82, 56, female). |
| | "The fact that I can actually always choose for myself, that I have the freedom to decide… even if we couldn't see each… it was actually always very accessible … if something was wrong, I could always write to her" (UKE105, 34, female). |
| | When there was a lockdown [the PSW] would call me and tell me every time, reminded me. She knows my memory is not excellent so she would call me and remind me 'next week you have the HEVRUTA [group PS meeting] do not forget' (BGU36, 37, female). |
| *Theme 2: The 'active ingredients' in the process of the intervention* | I do not necessarily need an expert. I just need someone who understands me and has gone through the same thing, and can continue to stand on both feet, and that we can both be an example to each other in that point (ULM47, 24, male). |
| | I think that the biggest point was this feeling that you have someone with whom you can talk about your problems, who knows that, who has also gone through that so. That is somehow something completely different than if you have had someone who has perhaps studied this or works in the job and does not know it. That was actually the most beautiful aspect for me (UKE79, 26, female). |
| | At first, I did not know that coming here to the UPSIDES peer support program would add value to my life. I used to feel that I am the only sick person, I am the only one who has problems without cure and my life was wasted. But when I was introduced to the UPSIDES peer support intervention I saw that there are other people who were older than me both men and women who were constantly going through mental health challenges (DS77, 29, female). |
| | The first intervention was about bringing me home, leaving me with medicines …, and they would not come back. But with UPSIDES, my PSW was coming frequently and this gave me a chance to share with him my challenges (BU70, 26, male). |
| | They [the PSWs] listened. Really listened… really tried getting inside our mind… but you know, not forcibly, not forcibly, would ask if you are ready to answer that 'are you ready? If you are not ready then just say no'… they did not force us to speak, they have listened to us… their attitude, their tolerance, their concern, the fun, they tried to have such fun meetings like this, you see what, they, they aren't robots, they are human beings, and I learned from them how to be a human being (BGU139, 58, female). |
| | There is a difference. The doctor measures our blood pressure, nurse gives us medicine. [the PSW] she just asks things like you do… her nature and attitude is really good. We sit and talk nicely with one another (PU40, 38, female). |
| *Theme 3: Positive outcomes of the intervention* | UPSIDES just got in the smallest threads of mental rehabilitation, where I had the greatest pain, where I understood myself the least and where I felt outside the circle… especially that the peers were copers [the term locally used for persons with mental illnesses] themselves and they could demonstrate to us how they have come from a place of suffering and they continue their life with contribution and a norm of support to the public… this is a fact that I was missing a lot and they just illustrated it to me and I started to believe I am also like them… it's unbelievable that in 12 meetings you can come out a different person. I started sharing and I got better medicine (BGU46, 36, male). |
| | I always thought I was like that, that no one can stand me… nobody wants to have anything to do with me. From time to time [the UPSIDES intervention sessions] it gave me a little more joy in life, that you go out and talk to her. And then maybe that infected me a little bit, that I've been doing so much lately, actually, in the last few months, that I have not done in a long time" (UKE95, 56, female). |
| | I was just at home expecting to be taken care of but after enrolling in this program I have become an entrepreneur, I do business, I take care of my child, I take care of myself. In the past I used to be in a state of dirtiness, but now I have learnt to have self-care, I have become an entrepreneur, I sell cleaning detergents, this program has helped me a lot … (DS111, 31, female). |
| | "My interaction with neighbors improved, I started doing some petty jobs around…, and they freely give me jobs without fearing me as a mad person (BU70, 26, male)." |
| | I can start with the positive effects that I have seen in the UPSIDES program, it has brought me close to my colleagues and the community, it has revived my dreams and given me hope in my mental health. (DS77, 29, female). |

**Table 2.** (*Continued*)

| Theme | Quotes |
|---|---|
| *Theme 4: Barriers* | The contact broke off so, that was just not great. I also wrote her roughly that I'm in the hospital, that I'm in inpatient … then suddenly, abruptly, yes, no more contact… so I had just wished that from my peer counselor, since she knew that I was in - in the hospital, maybe also from herself maybe again some feedback or questions or something. Or along the lines of [asking me] 'when are you coming out? And does it work to have a meeting or something?' … and at some point I said to myself: why should I run after them? … I found it difficult to stay in contact, because the thread between us was broken (UKE102, 38, female). |
| | Actually, during his first visit, I somehow felt bad. I was like…ha! [the hospital] people have followed me up to home? I just felt it within myself…, but afterwards I learnt to accept it …, because for sure who would help me if it's not the hospital! At first I thought it was violating my freedom yet I am already discharged from the hospital (BU70, 26, male). |
| | It caused me stress. Just the fact that I was coming to city X. I thought that someone would at least come to where I could take the bus myself. That would have been a better solution for me, but it did not work out… I had the feeling at some point that I was going there to fulfill my duty, but not because it was useful to me… and the tips, sure, he tried… but I already knew all that. I was already much further along. So, I do not need these beginner tips anymore, where I can get help… he just wasn't the right person for me. I do not want to devalue him, but for me it just wasn't the right person" (ULM72, 53, male). |
| | It was a challenge for me as I started to think about how I will go to the hospital… and I had to use my personal money. I faced challenges with sleep and woke up late, there was also a challenge with transport" (DS114, 24, male). |

Participant identifiers in quotations indicate the study site (BU = Butabika, Uganda; BGU = Be'er Sheva, Israel; DS = Dar es Salaam, Tanzania; PU = Pune, India; UKE = Hamburg, Germany; ULM = Ulm/Günzburg, Germany) followed by the participant number, age and gender.

and locations (hospital, community service, nature, *etc.*). This theme reflects service users' subjective experiences of the intervention's flexibility, allowing it to be tailored to the service users' needs and preferences, reinforcing their sense of being seen, cared for, and having choice.

Adaptability operated at two levels: At the site level, it enabled adaptation to local organizational and cultural characteristics (*e.g.*, a psychiatric hospital in Pune *versus* a community mental health service in Israel). At the intervention level, it allowed customization to the needs and preferences of PSW-Service-user dyads. Different locations and activities (*e.g.*, nature walks, wall-climbing halls) portrayed a room for creativity in the relationship between the PSWs and the service users, and many excerpts, mostly in low-income countries, described the comfort of having meetings at the service user's home. Flexibility also fostered service users' sense of autonomy:

> I also liked the fact that [PSW] didn't insist that we make more frequent appointments … so I wasn't under any pressure to see him frequently, but we could do it at a rhythm that suited me (UKE42, 52, female).

At the same time, it was evident that in sites where the intervention was structured such as the 12-sessions group meetings in Israel, an internal sense of autonomy was still preserved to some extent. In Israel, along with structure in terms of participant format (group meetings), setting (a single community mental health service), and the number of meetings (12), an expression of choice and autonomy was also presented:

> They asked us what kind of topics we want so it came also from us … we also wrote the rules of the group (BGU119, 27, male).

Flexibility ensured intervention continuity amid challenges, with phone contact providing a safety net and reliable support:

> He has not been coming but still he calls me and we talk, he even calls my mother to ask my wellbeing (BU61, 26, male).

This aspect became particularly important during the COVID-19 pandemic, mitigating its potential negative impacts by arranging outdoor meetings:

> There was the whole Corona theme and so … you just say, okay, we go for a walk in the fresh air somehow and just talk (UKE33, 33, female).

Overall, flexibility and the availability of the PSWs ensured autonomy and continuity. Beyond its practical benefits, flexibility appeared to function as an "active ingredient" in its own right, fostering satisfaction, perseverance, and the opportunity for service users to derive meaningful benefits.

### Theme 2: The 'active ingredients' in the process of the intervention

The second theme is composed of the core elements that service users experienced as 'active ingredients' that made the intervention influential, suggesting the mechanisms of impact. Depictions echoed the core values of peer support such as having a role model of recovery, reciprocity, reducing stigma, experiential knowledge, self-directed progression and more.

At the very basic level, service users felt seen and supported by trusted PSWs across sites:

> It's kind of nice when you know there's someone who takes you by the hand a bit and yes, as I said, you're then just not completely on your own … so if I hadn't had the peer support, I don't think I would be doing as well as I am right now (UKE105, 34, female).

Another participant vividly described the progressive process of relieving the burden of mental illness throughout the intervention:

> From meeting to meeting, as if there was a pile of sand and there's a truck that comes every day and unloads the pile … like a big mountain of mental problems and they came and took a little more and little more until at the end nothing was left, they just crumbled all this mountain of mental suffering and made it into nothing … (BGU46, 33, male).

Service users described a caring, mutual and reciprocal connection with PSWs. This space of trust allowed them to express their inner feelings and voices, promoting relief and decreased burden:

> I shared everything that I wanted to and I also cried a lot… and that really helped me feel light (PU61, 40, female).

Service users received words of advice and tools from PSWs, role-modeling recovery and encouraging to take action and responsibility, make choices, find inner strength, and progress on their personal path to recovery. PSWs modeled recovery, fostering hope:

> When I came to know that he is also a service user, I gained hope that I can also live with my mental illness. He encouraged me to keep taking medication… that I will be fine and be like him. He also encouraged me not to worry, to look for what to do…and that's how I got the courage to look for a job (BU109, 28, male).

Another key experience, fundamental to the intervention's impact, was the opportunity to foster human connection through it. While the experience of social inclusion can be regarded as an outcome, it was also a foundational element of the intervention itself:

> I used to be a person who isolated myself, but through the UPSIDES program I meet with other people who are also not well, so when we sit together and tell stories, ask each other questions, support and encourage each other you also become encouraged (DS77, 29, female).

The above quote demonstrates the centrality of the relational component provided in the process of the intervention.

### Theme 3: Positive outcomes of the intervention

Participants mentioned many positive outcomes resulting from UPSIDES, extending across various aspects of their lives. Gains included intra-personal improvements, such as relief from emotional burdens, decreased loneliness, higher self-esteem and higher confidence in managing emotions; inter-personal benefits, such as enhanced social skills, stronger community connections, and improved relationships with family; and behavioral changes, such as better self-care habits and increased adherence to treatment.

Service users frequently described emotional benefits using terms like 'happy,' 'light,' and 'less tense'. As mentioned by a service-user from Pune:

> When I let things out I feel really good and my mind also feels at peace (PU61, 40, female).

Another service-user portrayed the feeling of having a safe harbor in an unstable environment:

> So those who have problems… would like to have a foothold in life… a harbour… something that is safe. To have something secure in life, yes, that was also peer support for me (ULM47, 24, male).

Service users reported stronger social ties, inclusion and confidence, allowing self-expression and self-advocacy in circles outside the peer support meetings. Positive influences of the intervention on relationships with their families and within society were also shared. A service user shared that before UPSIDES, he felt ashamed to walk in his neighborhood. Through the intervention, he gained self-acceptance, family support, and ultimately reconnected with his daughter:

> I met my daughter, all of my family came to the party, they supported me and made me happy and it was a very big event of inclusion for me in society and suddenly I did not feel isolated anymore (BGU46, 33, male).

Behavioral aspects included self-care, treatment adherence, daily activities like hygiene, cooking, eating well, employment, mental health maintenance, and relapse prevention:

> Since I started to receive UPSIDES peer support, I have never been sick… before being enrolled in this program I used to experience frequent relapses" (DS111, 31, female).

Along with adopting positive behaviors, some participants reduced harmful habits like substance abuse:

> I don't drink anymore… my head, my health. I don't want this anymore. I don't need this anymore. And I won't stay with one [who drinks]" (ULM84, 20, male).

Some variation in how medication adherence was described emerged across sites. In middle- and low-income countries, adherence to psychiatric medication was frequently emphasized. Service users stated that PSWs actively encouraged them to take their medication regularly. The interviews suggest that this was not perceived as controlling but rather as supportive, together with other activities of self-care:

> Through the UPSIDES intervention, I have been able to identify disease symptoms and manage the symptoms such as through taking medication and psychological therapy… as days go by I need to know the symptoms, treatment, and how to live with the problem without giving up (DS77, 29, female).

In high-income countries like Israel, service users focused on treatment autonomy and re-evaluating medication:

> The topic of responsible reducing [i.e. tapering psychiatric medication] came to mind and I said, maybe I can talk to him [the psychiatrist] about reducing the dosage… I said there is a whole world that I can [pause] that I can do more research about… It made me feel more optimistic about the future, [because] there are other options (BGU119, 27, male).

Whilst adhering to psychiatric medication and tapering doses may seem like opposing actions, both represent a form of self-directed behavior and autonomy.

### Theme 4: Barriers

Overall, the breadth of barriers mentioned was relatively low in comparison to the positive aspects, and strictly negative outcomes of the intervention were not mentioned, despite being directly inquired about. Nonetheless, service users highlighted potential shortcomings that imply suggestions for improvement.

While flexibility was valued, rigidity limited the intervention's adaptability. A few participants mentioned that their respective PSWs were not flexible enough in means of contact platforms or meeting locations. Similarly, where 'active ingredients' were lacking, the intervention's effectiveness was diminished. PSWs who did not adhere to the core value of reciprocity (from the service users' perspective), led to a feeling that the relationship is one-sided:

> I opened up completely, revealed myself, and she somehow didn't … only told me superficially that she lives alone and well, I don't know much more (UKE95, 56, female).

While flexibility was generally seen as a positive feature, it occasionally caused difficulties due to blurred boundaries, leaving room for negative interpretations. Unclear arrangements led to confusion and misunderstandings at various points: in the beginning of the intervention, during its course, or at its conclusion. At the start of the intervention, one participant found home visits intrusive, perceiving the PSW's outreach as an attempt to admit him:

> So I would feel like, they are trying to get me admitted (PU80, 39, male).

Similarly, the interventions' unclear ending could also lead to confusion:

My PSW said he will stop coming to visit and I don't know why… I felt bad because I thought that maybe I didn't treat him well…or maybe he is tired of me, little did I know that it was an instruction from the project (BU61, 26, male).

A discrepancy arose between service users' expectations of the peer support intervention and its reality, with some anticipating a more therapeutic relationship:

I had imagined it more like therapy with someone who has done it before. But of course, that's not the point of peer support (UKE79, 26, female).

Other instances of a personal mismatch between the service-user and the PSW hindered the establishment of a reciprocal relationship, such as a participant who felt more advanced than the PSW, finding the support unhelpful and difficult to engage with. When such feeling of mismatch occurred, the balance could be tilted from a reciprocal relationship to one in which the service-user felt obliged to attend, without gain:

Feeling like I have to hold the space for the other person (UKE33, 33, female).

This appears to be a unique experience for service users from high-income countries, such as Germany.

Another site-specific example for barriers was in Israel, where factors related to difficulties maintaining group attendance were mentioned:

Other people quit… there were two coming so it was a very very small group… it was less of an opportunity to make connections with people and that's a pity (BGU36, 37, female).

Lastly, geographical challenges made in-person meetings difficult due to distance, time, and cost:

if I would have lived closer it would have been easy but where I live currently, it is the last bus stop and commute is very difficult (PU61, 40, female).

This experience was not shared by many participants, but did appear in different sites therefore pointing to the matter of location and accessibility.

## Discussion

The study examined service users' experiences of the UPSIDES peer support intervention across diverse socio-economic, cultural, and organizational contexts. Four themes emerged: intervention flexibility; the active ingredients of peer support (*e.g.*, mutuality, reciprocity, and role-modeling recovery); positive intrapersonal, interpersonal, and behavioral outcomes; and barriers including mismatches between service users and PSWs, unmet expectations, unclear boundaries, and continuity challenges.

Service users' experiences reflect the consistent use of recovery-oriented concepts and benefits, including experiential knowledge, social inclusion, practical recovery strategies, positive role models, reduced self-stigma, and an enhanced sense of community belonging. These findings align with previous studies supporting the value of peer support (Gillard et al., 2015; Tse et al., 2017; Rosenberg and Argentzell, 2018; Marks et al., 2022) and also complement the quantitative findings reported in the UPSIDES trial (Puschner et al., 2025). Findings suggest that flexibility is essential to cross-cultural and person-centred MHPS implementation. In other words, in a similar manner to other interventions that include mentoring features (Lyons and McQuillin, 2021), heterogeneity is not a 'bug' resulting from uncontrolled circumstances, but a

'feature' central to the essence of peer support. Amid the flexibility, the consistency in relation to the 'active ingredients' across sites is notable. It appears that the combination of core values with flexibility, largely mitigated potential shortcomings of peer support identified in previous studies, such as rigidity in PSW approaches and institutional regulations (Ogundipe et al., 2019). Moreover, adverse effects or harm, previously suggested as potential concerns in peer support (Le et al., 2022) were not reported.

Although contextual differences were observed, experiential patterns converged more than they differed, suggesting that the core relational aspects of peer support may operate similarly across settings. This inference is particularly compelling in relation to LMICs, where mental health services face significant challenges, including limited resources (Mpango et al., 2020), and more stigmatizing attitudes toward mental illness (Ramesh et al., 2023).

Together with these shared experiences, some context-specific practices also emerged. Meetings at the service user's home were unique to LMIC sites, along with descriptions of interactions between the family and the service user. Expectations of peer support appeared to be more specific in high-income countries (*e.g.* expecting the PSW to match in recovery stage), while in settings where mental health services were rare, participants frequently emphasized attention, empathy, and companionship. Supportive human connection itself appeared highly valued. A stronger focus on encouraging service users to adhere to psychiatric medication was also observed in LMIC sites in contrast to participants in Israel, a high-income setting, who described considering tapering psychiatric medication. Similar patterns were reported in a study exploring mental-health workers' perspectives on peer support (Krumm et al., 2022). Lastly, improvements in daily living and employment were reported more frequently in low-income settings.

These cross-site patterns may relate to differences in mental health system structures and resource environments. In low-income settings, peer support appeared closely connected to access to care, illness management, and practical support, reflecting contexts where treatment resources are limited and family involvement often plays a central role (Morillo et al., 2022). In such settings, peer support may partly function as a bridge to limited resources and opportunities, amplifying gains less available through other services. Meanwhile, participants in high-income settings more often emphasized autonomy, identity development, and navigating recovery choices within established service systems. Adhering to medication or considering reduction may appear contradictory, but may represent different contextual expressions of the same principle of self-directed recovery. In resource-limited settings, supporting adherence may function as a stabilizing practice, whereas in more resource-rich systems autonomy may be enacted through greater emphasis on personal choice and re-evaluating treatment decisions (Asher et al., 2025). Peer support may promote autonomy by helping individuals make informed decisions within the constraints and opportunities of their local care systems.

Barriers highlight factors that may hinder the intervention's impact. Previously noted pitfalls of peer support, such as confusion, mistrust, or relational mismatch between peer workers and service users (Lewis and Foye, 2022; Ong et al., 2023) were also reflected to some extent in the current findings. This suggests a structured matching process to pair PSWs and service users based on geographical proximity and stage of recovery. Such alignment can enhance the quality of the relationship and foster mutual trust. Continued oversight, feedback mechanisms, and progress evaluations may also help identify disengagement early and prevent the intervention from

becoming burdensome. Unlike many forms of clinical care, the peer support contract involves a bidirectional process of boundary setting and adjustments (Knopes and Dégale-Flanagan, 2023). Therefore, key aspects of the intervention such as timeframe and duration, forms of communication, roles, responsibilities, and limitations of both parties should be explicitly conveyed. This may minimize confusion and help both sides navigate preferences within these boundaries.

### *Strengths, limitations, and future research directions*

This study captures service user perspectives on peer support across diverse cultural and socio-economic contexts. The mixed-methods sampling strategy ensured a broad range of experiences and enhanced analytical depth. Several limitations should be noted, reflecting both the contextual nature of qualitative research, and the design of the study. Although similar experiential patterns emerged across sites, the findings were generated within specific cultural and organizational contexts, and their interpretation should take these contexts into account. Furthermore, despite systematic translation processes, some cultural nuances may not have been fully preserved, a common challenge in multi-national research. With respect to the study design, in accordance with the trial protocol, interviews were conducted after the intervention concluded at all sites, creating a time gap that may have affected the accuracy and depth of recollections. Selection bias is possible, as only those who agreed to be interviewed were included, and the target of six interviews per site was not always met, potentially skewing findings toward more favorable experiences. Although 'low responders' were purposely sampled to address this, critical perspectives may still be underrepresented. Additionally, social desirability bias may have influenced responses, though this was mitigated by repeated inquiries about negative experiences. Finally, the three primary analysts were all from high-income countries. Sites in lower income countries contributed to the cross-site validation, and this study was undertaken in the context of a larger UPSIDES study which had developed strong sensitivity to cultural differences, by conducting a situation analysis (Ibrahim et al., 2020), co-developing a theory of change (Hiltensperger et al., 2024), cross-cultural intervention piloting (Nixdorf et al., 2022), investigating cultural influences on peer support worker implementation (Ramesh et al., 2023) and identifying wider societal perspectives on UPSIDES peer support (Haun et al., 2024). However, future research might compare the interpretative themes produced by primary analysts from higher *versus* lower income settings.

Although participants were purposely selected to include individuals with high and low responses to the intervention, the qualitative analysis did not reveal clear experiential differences between these groups. Including the perspectives of service users who discontinued the intervention in future studies could help further explore how participants' experiences relate to variation in outcomes, including less positive trajectories. Another area for future research concerns the role of flexibility in peer support implementation. Flexibility may be a useful lens for assessing and strengthening peer support interventions, especially models not originally designed for diverse cultural and service environments.

### Conclusion

The study underscores the universal essence and adaptability of peer support across diverse contexts, reinforcing its value as a complementary service independent of clinical care. The UPSIDES intervention illustrates how core peer support principles can be operationalized with flexibility, enhancing cross-cultural relevance and person-centred care. Challenges include service-user-PSW matching and clearer boundaries. Overall, the findings affirm peer support as a distinct, adaptable approach with significant global potential. By drawing on service-user perspectives across sites with differing resources and sociocultural norms, this study contributes new evidence on how peer support can be scaled and adapted globally, particularly in low- and middle-income settings where mental health systems remain under-resourced.

**Open peer review.** To view the open peer review materials for this article, please visit http://doi.org/10.1017/gmh.2026.10203.

**Supplementary material.** The supplementary material for this article can be found at http://doi.org/10.1017/gmh.2026.10203.

**Data availability statement.** Transcript fragments which informed the analysis presented in this publication are included within the paper. Full transcripts are not publicly available due to their containing information that could compromise the privacy of research participants.

**Acknowledgements.** The study Using Peer Support In Developing Empowering Mental Health Services (UPSIDES) is a multicentre collaboration between the Department for Psychiatry and Psychotherapy II at Ulm University, Germany (Bernd Puschner, coordinator); the Institute of Mental Health at University of Nottingham, UK (Mike Slade); the Department of Psychiatry at University Hospital Hamburg-Eppendorf, Germany (Candelaria Mahlke); Butabika National Referral Hospital, Uganda (Juliet Nakku); the Centre for Global Mental Health at London School of Hygiene and Tropical Medicine, UK (Grace Ryan); Ifakara Health Institute, Dar es Salaam, Tanzania (Donat Shamba); the Department of Social Work at Ben Gurion University of the Negev, Be'er Sheva, Israel (Galia Moran); and the Centre for Mental Health Law and Policy, Pune, India (Jasmine Kalha). We thank Philip Wolf for his essential role in the data procedures, which enabled the mixed-methods design of this study.

**Author contribution.** Conceptualization and study design were led by S.K., B.P., and G.M. A.G., M.H., S.K., and G.M. developed the interview topic guide. All authors contributed to the work through their involvement in the data acquisition process. Data analysis was conducted by Y.G., A.G., and G.M. The original draft of the manuscript was prepared by Y.G. and critically reviewed by A.G. and G.M. for important intellectual content. The manuscript was critically revised for important intellectual content by all authors (Y.G., A.G., S.B., C.H., J.K., R.M., R.N., B.P., M.S., M.H., S.K., and G.M.). All authors approved the final version for publication and agree to be accountable for all aspects of the work, ensuring that questions related to the accuracy or integrity of any part of the work are appropriately investigated and resolved.

**Financial support.** UPSIDES has received funding from the European Union's Horizon 2020 research and innovation programme under grant agreement No 779263. This publication reflects only the authors' view. The Commission is not responsible for any use that may be made of the information it contains.

**Competing interest.** All authors declare that they have no conflicts of interest.

**Ethics statements.** The authors assert that all procedures contributing to this work comply with the ethical standards of the relevant national and institutional committees on human experimentation and with the Helsinki Declaration of 1975, as revised in 2013. All procedures involving human subjects/patients were approved by: Ethics Commission of Ulm University, Germany (ref. 254/19), the Local Psychological Ethics Commission at the Centre for Psychosocial Medicine, Hamburg, Germany (ref. LPEK-0095), the Uganda National Council for Science and Technology (ref. SS 4990), the National Institute for Medical Research, Dar es Salaam, and Ministry of Health, Community Development, Gender, Elderly & Children, Dodoma, Tanzania (ref. NIMR/HQ/R.8a/Vol. IX/3328), the Human Subjects Research Committee of Ben-Gurion University, Israel (ref. 1787–1) and the Indian Law Society (ref. ILS/37/2018). Informed

consent was obtained from all participants, and data were anonymized in compliance with the General Data Protection Regulation (GDPR) and relevant local data protection laws.

**Transparency declaration.** The lead authors (YG and GM: manuscript guarantors) affirm that this manuscript represents an honest, accurate, and transparent account of the study being reported. No important aspects of the study have been omitted, and any discrepancies from the study as initially planned have been fully disclosed and explained.

**Use of artificial intelligence (AI) tools.** We used a large language model (LLM; ChatGPT, OpenAI GPT-5.1) to assist with general language editing, including improving clarity, readability, and grammar. The tool was not used to generate scientific content, ideas, data analysis and interpretations. All AI-assisted text was reviewed, checked, and edited by the authors. The authors take full responsibility for the accuracy and integrity of the manuscript.

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
