## [Reviewer Report]

The authors present a multi-country qualitative study exploring service users’ experiences with UPSIDES peer support across high-, middle-, and low-income settings. Using semi-structured interviews (n=33), they identify four themes: adaptability/flexibility, “active ingredients,” positive outcomes, and barriers. The paper aims to demonstrate the universal value of peer support while capturing contextual differences relevant to global implementation.

This is a well-written, methodologically transparent, and timely manuscript with strong relevance for global mental health and peer support implementation. However, several conceptual, methodological, and analytical issues need clarification before publication.

1. Qualitative research is inherently contextual, yet the paper frequently treats findings as universal. Given the vast cultural, organisational, and linguistic differences across sites, the authors should clearly articulate why such a generalisation is valid. Do acknowledge the limit of transferability.

2. The analysis does not deeply explore cultural differences. Important distinctions - such as the meaning/interpretation of home visits, stigma patterns, family involvement, or local psychiatric norms - do not fully surface. Moreover, the coding team appears predominantly based in high-income settings. Do try and involve the perspective of researchers from LMICs. Provide richer comparative insight rather than presenting mostly global summaries.

3. Several themes (e.g., positive outcomes, barriers) function more as broad categories than interpretive concepts. A more interpretive voice is needed, which can focus on questions like - why was medication adherence emphasised in LMIC settings, whereas autonomy/tapering dominated in HICs?

4. The paper identifies “flexibility” as both strength and challenge. However, the negative experiences described (e.g., unclear endings, confusion about home visits, mismatched expectations) point more towards lack of communication and unclear boundaries, rather than flexibility per se. Reframing would be helpful.

5. The sampling strategy intentionally contrasted “high” vs. “low” responders based on quantitative outcomes, yet the authors report no notable differences in experiences. The authors should discuss the reasons and implications of the same.

---

## [Reviewer Report]

General comments

This paper is a welcome contribution to the literature on service user experiences of mental health services, particularly as it is tied to a larger project. I have some suggestions for revision, to improve clarity and accessibility. I appreciate that the study is part of a larger project which has been reported elsewhere and may be familiar to some, but not all, readers. Therefore some of my points indicate a need to share information about the project as a whole, to make sense of your decisions for this study.

A decision is needed in the writing team about the terms you are using to describe your participants/service users/clients. If the decision has already been to vary these terms, to reflect the international context, then that could be explained in the introduction/background. If not, then it might be appropriate to discuss revision, to ensure the consistent use of these terms across different sections.

It would be helpful to understand better who has done what. In particular:

The local research teams: were they recruited for the UPSIDES project as a whole? Were they considers part of the UPSIDES research team, or working on that team’s behalf?

The UPSIDES research team: what roles did team members have apart from translators?

Where did the moderators, interviewers, bilingual speakers come from?

Method

It could be helpful for readers unfamiliar with UPSIDES to briefly indicate the findings of the randomised controlled trial and related studies, to strengthen your argument for investigating service user experiences. Was it broadly successful and if so, in which respects? It is particularly interesting that there are common themes across different international and cultural contexts, if an intervention is flexible enough.

P4, line 7: “meetings were delivered” - while this wording reflects the language of an intervention study, in the context of this paper with its focus on service user experiences it could be reworded. to avoid positioning service users as passive recipients of a package.

P5, line 7: it is interesting that content analysis revealed little difference between the two groups of participants. More detail about that analysis is required.

It might help to link the countries involved with the acronyms in the quotes BU, BGU, DS, PU, UKE, ULM, as there appear to be six locations but only five countries for formal approval.

Results

P5, line 38 what evidence is there to support the claim that it was the PSW’s creativity, and not a joint effort, to identify locations and activities?

P5, line 45 “the intervention was mostly structured” - what does this mean and how does it differ from the way the intervention has already been described in the method section?

P9, line 22 the use of the word “unique” is confusing and this point could be clarified, if it is important. Is the implication that expectations of peer support are different in high income countries? If so, how is that unique?

P9 line 27 were any of your participants people that did not complete the UPSIDES intervention? It might be helpful in the brief summary of the intervention to indicate how many did drop out and maybe whether there were differences between the group and pair versions.

Discussion

P9, line 43 It might be helpful at this stage to reiterate the themes and the “active ingredients” to make it clearer what you mean by “mechanisms of impact”

P10, line 38 how do the key aspects relate to the active ingredients, given the latter’s importance in the paper so far?

P10, line 55 the point about social desirability made me wonder if the interviewers for this study were involved in recruiting for the intervention or overseeing it in some way? It may be helpful to indicate somewhere if they were not.

Conclusion

P11, line 13 I do wonder if peer support is an approach, rather than a model, as you have stated that flexibility is such an important aspect of it. My view here reflects the usefulness of defining recovery as an approach rather than a model in many clinical settings, giving scope for adaptation to local resources and opportunities.

---

## [Reviewer Report]

This manuscript presents qualitative findings from the international UPSIDES trial examining service users’ experiences of peer support in mental health services. The topic is important and timely, and the inclusion of multi-country qualitative data has the potential to make a meaningful contribution to the peer support literature. However, the manuscript would benefit from substantial revisions to improve clarity, completeness, and conceptual alignment between the study aims, methods, results, and framing of user experiences. In its current form, the paper is underdeveloped in several key sections—particularly the abstract, introduction, and methods—which limits the reader’s ability to fully understand the study design, analytic approach, and contribution to the literature.

Abstract and Impact Statement:

• The abstract is quite brief, particularly with regard to the methods. While sites and purposive selection are mentioned, there is insufficient detail to understand how the qualitative study was conducted. The authors should expand the methods portion of the abstract to include: that semi-structured interviews were conducted and the general analytic approach used (e.g., thematic analysis).

Introduction:

• The introduction is very brief and does not effectively establish a clear gap in the existing literature. While the two introductory paragraphs outline peer support in mental health services, they do not sufficiently articulate what is not yet known, particularly regarding service users' experiences of peer support across international contexts.

• The second paragraph reviews relevant literature, but some sentences are difficult to follow and would benefit from clearer wording and tighter structure.

• Overall, the introduction could remain concise while more clearly articulating gaps in qualitative research on peer support, explaining why international, cross-site qualitative data are needed, and clearly stating what this study adds beyond existing work.

Methods:

The Methods section would benefit from some expansion and clarification.

1. Description of the Parent Trial

a. The sentence stating that the project included a randomized controlled trial “employing a mixed methods evaluation” is confusing and grammatically awkward. “Employing a mixed methods evaluation” functions as a dependent clause and should be reworded.

b. More information should be provided about the parent trial, including a lengthier description of the UPSIDES intervention and the goals and clinical outcomes (given they guide participant selection). I would also consider adding a separate measures section within this manuscript to more fully describe the measures.

2. Qualitative Methods

a. The authors reference COREQ guidelines but never define the acronym.

b. Some detail is needed on the semi-structured interview guide. At a minimum, a brief description of the key domains covered and whether it was adapted across sites should be included.

Results:

The qualitative results are generally well written, and participant quotations are appropriately and effectively integrated.

• Alignment Between Results and Study Focus

o While the themes are clearly presented and well-supported by quotes, many appear to focus more on intervention implementation or the broader UPSIDES program rather than on service users’ lived experiences of peer support specifically.

o This is a nuanced issue, but it raises the question of whether the manuscript title accurately reflects the primary focus of the findings. The authors may wish to consider whether the title should be revised to better align with the content.

• Participant Identifiers

o If the participant ID codes (letters preceding numbers) correspond to study sites or other characteristics, a key should be provided so readers can better contextualize quotations.

• Site-Level Variation

o The manuscript notes “notable variation across sites” regarding medication adherence. Unless the authors conducted a specific analytic approach to examine cross-site differences, such statements may overreach and should be either softened or removed.

Discussion:

• The discussion engages relevant literature and appropriately situates findings within broader conversations about peer support; however, it would benefit from careful editing to improve clarity and readability.

o Some sentences and paragraphs would benefit from tightening to ensure they are easily interpretable.

Minor Comments:

1. The acronym “SU” is used but never defined and should be clarified upon first use.

2. The first sentence of the Results section is not a complete sentence and should be revised.

---

## [Reviewer Report]

Very concisely written and interesting article that is an important contribution to the developing field of Peer Support. Just a couple of brief suggestions that might make the article more clear for readers with experience of implementing Peer Support.

1 - s.3/22 Peer(s)?

2 - Background - It would be interesting to know more about the organisational context for the various Upsides sites. While table 2 provides basic information readers may be curious to hear a bit more about resources, peer support employment conditions (all paid? Training? Volunteers? Integrated in MH teams? etc. This just to concretize the implications of implementing peer support in low and middle income countries, but even high income (the comparative aspect inherent in the study).

3) Some more background to potential cultural differences between these countries/sites would also help the reader to understand the dimensions of the study and the significance of the results, that the mechanisms seem to be the same across high, low, middle....

4 - The Upsides intervention (under method) could be more clearly described. The lack of clarity as to the actual intervention is added to by the variance in delivering it one-on-one or group, and may lead to questions as to whether this variance undermines the study results. I don´t see that, but good to summarize the intervention in relation to the core principles.

The discussion and conclusions are well written and compelling. The essential nature of flexibility in relation to the challenges in low income countries, might suggest a need for further research into cultural adaptations appropriate in varied settings, Thank you for the opportunity to read this interesting paper.

---

## [Editor Report]

Dear Authors 

We have received reviewer comments and our recommendation is major revision to the manuscript. 

Only then can it be considered for publication 

Regards

Siham

---

## [Reviewer Report]

The authors have addressed most comments well. I have a few remaining points for clarification.

The concept of “adaptable settings” appears to include differences such as hospital versus community-based delivery. However, these variations may reflect site-level or researcher-driven decisions rather than service user choice. Could the authors clarify how this constitutes flexibility from the service user’s perspective?

The term “active ingredients” suggests clearly defined mechanisms of change within the intervention. However, the elements listed (e.g., role modelling, reciprocity, feeling supported) appear broad and potentially overlapping. It would be helpful for the authors to more clearly define what qualifies as an “active ingredient.”

The reintroduction of “flexibility” in the context of barriers (page 11, line 24) raises some conceptual ambiguity. It is unclear whether the issue lies in excessive flexibility, poorly structured flexibility, or lack of clarity around boundaries. Greater differentiation would help avoid conflating flexibility as both a strength and a limitation.

While the authors integrate contextual differences into their interpretation of findings, the contextual nature of qualitative research is also described as a limitation (page 14, line 24). Could they clarify this positioning—specifically, in what sense context constrains the findings despite being analytically engaged with?

---

## [Reviewer Report]

I think the authors have done an excellent job of working with and responding to the recommendations. The article is now ready for publication.

---

## [Editor Report]

Dear Authors 

We have received reviewers comments on the revised version of your submitted manuscript.

Pleased to accept the manuscript for publication, we will inform you of the next steps. 

Regards

Siham